# Evaluation of transabdominal and transperineal ultrasound-derived prostate specific antigen (PSA) density and clinical utility compared to MRI prostate volumes: A feasibility study

**Maria Pantelidou[1], Iztok Caglic[1], Anne George[2], Oleg Blyuss[3,4,5], Vincent J. Gnanapragasam[2,6,7], Tristan Barrett[1] ***

1 Department of Radiology, Addenbrooke's Hospital, University of Cambridge School of Clinical Medicine, Cambridge Biomedical Campus, Cambridge, United Kingdom, 2 Cambridge Urology Translational Research and Clinical Trials Office, University of Cambridge, Cambridge, United Kingdom, 3 School of Physics, Engineering & Computer Science, University of Hertfordshire, Hatfield, United Kingdom, 4 Department of Paediatrics and Paediatric Infectious Diseases, Sechenov First Moscow State Medical University, Moscow, Russia, 5 Department of Applied Mathematics, Lobachevsky State University of Nizhny Novgorod, Nizhny Novgorod, Russia, 6 Division of Urology, Department of Surgery, University of Cambridge, Cambridge, United Kingdom, 7 Department of Urology, Addenbrooke's Hospital, Cambridge Biomedical Campus, Cambridge, United Kingdom

* tristan.barrett@addenbrookes.nhs.uk

**Data Availability Statement:** We have uploaded the dataset file on https://data.mendeley.com/, with the appropriate original data file relevant to this

## Abstract

### Purpose

To investigate the accuracy of surface-based ultrasound-derived PSA-density (US-PSAD) versus gold-standard MRI-PSAD as a risk-stratification tool.

### Methods

Single-centre prospective study of patients undergoing MRI for suspected prostate cancer (PCa). Four combinations of US-volumes were calculated using transperineal (TP) and transabdominal (TA) views, with triplanar measurements to calculate volume and US-PSAD. Intra-class correlation coefficient (ICC) was used to compare US and MRI volumes. Categorical comparison of MRI-PSAD and US-PSAD was performed at PSAD cut-offs <0.15, 0.15–0.20, and >0.20 ng/mL$^2$ to assess agreement with MRI-PSAD risk-stratification decisions.

### Results

64 men were investigated, mean age 69 years and PSA 7.0 ng/mL. 36/64 had biopsy-confirmed prostate cancer (18 Gleason 3+3, 18 Gleason ≥3+4). Mean MRI-derived gland volume was 60 mL, compared to 56 mL for TA-US, and 65 mL TP-US. ICC demonstrated good agreement for all US volumes with MRI, with highest agreement for transabdominal US, followed by combined TA/TP volumes. Risk-stratification decisions to biopsy showed

study (DOI:10.17632/2bvdcspxx2.1); it can be accessed via https://data.mendeley.com/datasets/2bvdcspxx2/1.

**Funding:** This research was supported by the National Institute of Health Research Cambridge Biomedical Research Centre (BRC-1215-20014). The views expressed are those of the author(s) and not necessarily those of the NIHR or the Department of Health and Social Care. The authors also acknowledge support from Cancer Research UK (Cambridge Imaging Centre grant number C197/A16465), the Engineering and Physical Sciences Research Council Imaging Centre in Cambridge and Manchester and the Cambridge Experimental Cancer Medicine Centre. The funders had no role in study design, data collection and analysis, decision to publish, or preparation of the manuscript.

**Competing interests:** The authors have declared that no competing interests exist.

concordant agreement between triplanar MRI-PSAD and ultrasound-PSAD in 86–91% and 92–95% at PSAD thresholds of >0.15 ng/mL$^2$ and >0.12 ng/mL$^2$, respectively. Decision to biopsy at threshold >0.12 ng/mL$^2$, demonstrated sensitivity ranges of 81–100%, specificity 85–100%, PPV 86–100% and NPV 83–100%. Transabdominal US provided optimal sensitivity of 100% for this clinical decision, with specificity 85%, and transperineal US provided optimal specificity of 100%, with sensitivity 87%.

## Conclusion

Transperineal-US and combined TA-TP US-derived PSA density values compare well with standard MRI-derived values and could be used to provide accurate PSAD at presentation and inform the need for further investigations.

## Introduction

Prostate cancer (PCa) accounts for almost 1-in-5 of all new male cancer diagnoses [1], with the incidence of the disease expected to double by 2030 in part due to the ageing population [2]. The adoption of prostate-specific antigen (PSA) test for screening symptomatic patients in the mid-1990s dramatically changed the profile of PCa patients, with trends towards detection of lower grade disease, including clinically insignificant tumours [3]. A raised PSA level only has a 30% positive predictive value for cancer [4] and is more commonly associated with benign prostatic hypertrophy (BPH), conversely, 37% of men with a PSA reading in the normal range of 2.5–4 ng/ml still harbour prostate cancer [5], thus PSA screening is not routinely recommended [6]. Prostate multiparametric (mp) MRI can aid the diagnostic process by avoiding biopsy when negative in low-risk patients and in directing biopsy to suspicious targets when positive, and is now recommended within international and European guidelines as the initial diagnostic step for men with suspected PCa [7, 8].

MRI has a high negative predictive value for the presence of clinically significant (cs) PCa and can help avoid biopsy in 27–49% of patients [9–12]. However, MRI is an expensive resource and there is increasing pressure on MRI services to provide rapid access to MRI and prompt reporting times, in the face of ever increasing demand [13]. Given that a high proportion of mpMRI studies are negative, a triage step that could safely avoid such imaging in a proportion of patients would be desirable. PSA-density based on MRI-derived prostate volumes has been shown to be a more reliable biomarker than PSA alone [14–16]. In addition, PSA-density is now considered an integral part of the biopsy decision making process, with a recent consensus paper stating that PSA-density should be used to help avoid biopsy when MRI is negative and to augment the decision-making process when MRI findings are equivocal [17]. However, prostate gland volume can also be derived from other imaging modalities including ultrasound (US) which, although operator-dependent, is relatively inexpensive, quick to perform, and widely available. To date, however, most studies comparing US with MRI have employed transrectal ultrasound which while is invasive and intrusive for patients. A more acceptable alternative is surface-based US with images acquired via either the abdomen or perineum.

The aim of this study was therefore to investigate how well surface-based US-derived PSA-density performs in comparison to the gold-standard of MRI-derived PSA-density and

whether it could be used in clinical practice as a risk-stratification and triage tool for deciding whether patients require urgent MR imaging or could safely avoid further investigations.

## Methods

This was a single tertiary centre prospective cohort feasibility study, with written informed consent obtained from all participants. All elements of this prospective study were carried out in accordance with the Declaration of Helsinki and were approved by the institutional ethics board (NRES Committee East of England, UK, ref: NRES 03/018), with written informed consent obtained from all participants. All methods were performed in accordance with the relevant guidelines and regulations. The inclusion criteria for the study were men having a prostate MRI for suspected prostate cancer and subsequently undergoing prostatic biopsy, regardless of whether cancer was present on biopsy. Exclusion criteria were a clinical diagnosis of prostatitis and/or abscess, presence of hip metalwork, and/or any previous treatment for prostate cancer. 65 participants were recruited, with one patient excluded due to loss of imaging data transfer to PACS. The primary outcome of the study was to compare US-derived prostate volume with MRI prostate volume and thus determine the accuracy of surface-based US-derived PSA-density values. The secondary outcome was to derive relevant US PSA-density cut-off values that could be applied in clinical practice to predict the need for biopsy and/or presence of significant PCa.

### Ultrasound imaging

All scans were performed on a standard commercially available ultrasound machine (Toshiba Aplio XG), using a low frequency curvilinear abdominal transducer. All patients were scanned with a full-bladder by a single, experienced uro-radiologist. Two standard ultrasound views were performed: trans-perineal (TP) and trans-abdominal (TA). For each view, three orthogonal measurements were acquired and used to calculated the US-derived prostatic volume: right-to-left (axial 1), antero-posterior (axial 2) and cranio-caudal (sagittal) (**Fig 1**).

### MRI imaging

3T-MRI (MR750, GE Healthcare) was performed for all participants using a 32-channel phased-array coil. The following sequences were acquired: T2-weighted 2D fast recovery FSE images in axial, sagittal and coronal planes and sagittal T2 3D FSE sequences, axial T1-weighted imaging: TR/TE 561/11 ms, FOV 24 × 24 cm, resolution 1.1 × 1.0 mm; axial diffusion-weighted imaging (DWI) dual spin echo-planar imaging and axial 3D dynamic contrast–enhanced (DCE) imaging following a bolus of Gadobutrol (Gadovist, Bayer). The T2-weighted imaging was used to derive prostate volume data.

### Imaging analysis

Four different US-derived prostatic gland volumes were derived, using transabdominal (TA) and transperineal (TP) views: US-TA volume, US-TP volume, and using two combinations of TA and TP volumes, US-TA-TP volume and US-TP-TA, respectively, to overcome shadowing from the pubis bone (**Fig 2**). US TA-TP was derived using the Axial 1 and Axial 2 measurements of derived from transabdominal views and combined with the sagittal measurement acquired transperineally (**Table 1**). US-TP-TA was derived using the axial 1 transabdominal measurement combined with the Axial 2 and sagittal measurements acquired transperineally. MRI volume was calculated using triplanar measurements derived from the axial and sagittal T2-weighted sequences, following the PI-RADS version 2.1 guidelines ellipsoid formulation;

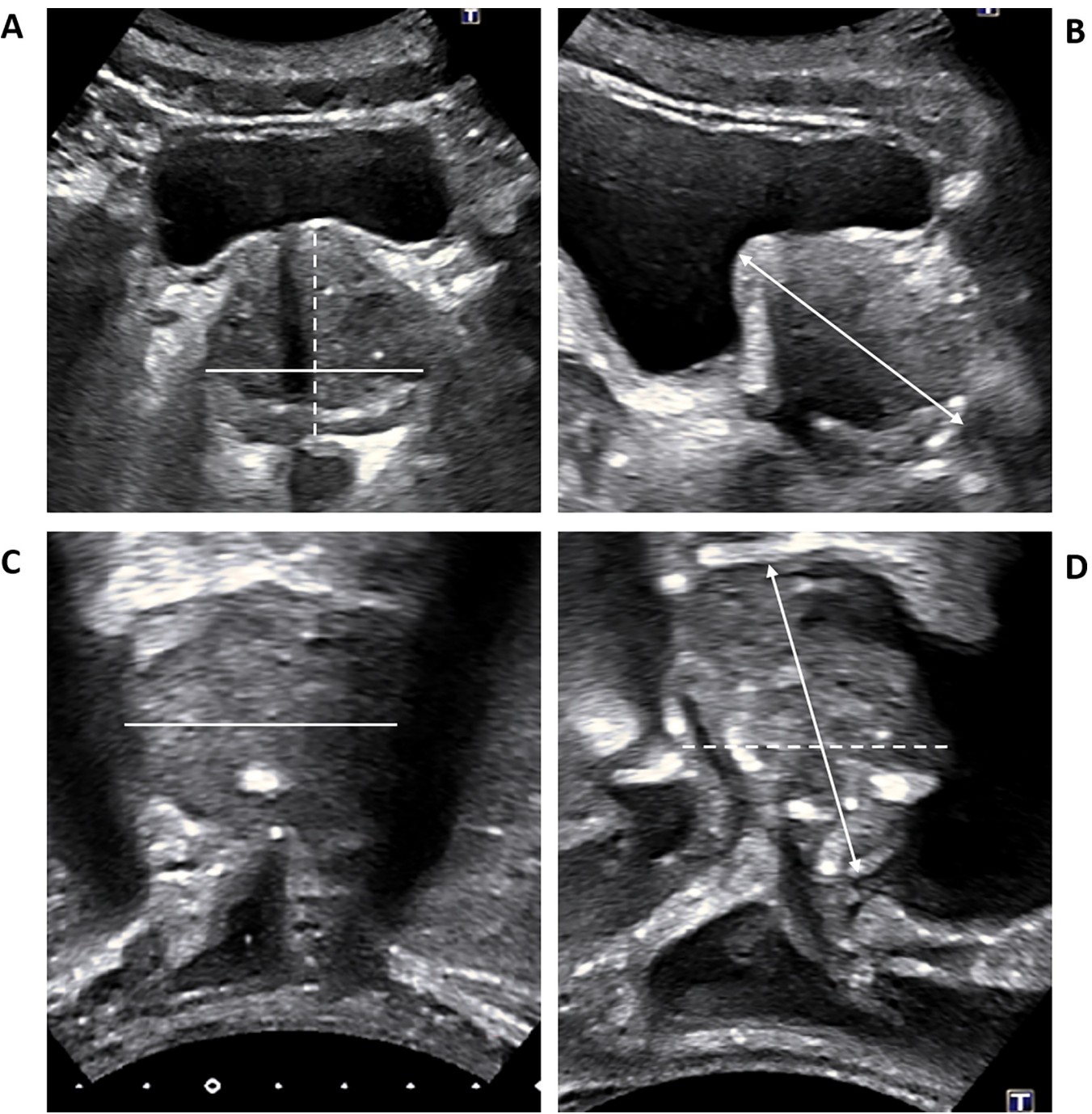

**Fig 1. Ultrasound views obtained.** Transabdominal views in the transverse/axial (A) and longitudinal/sagittal (B) planes. Transperineal views in the longitudinal/sagittal (C) and transverse/coronal planes (D). Lines represent "Axial 1" measurements (right to left), dashed lines represent "Axial 2" measurements (anterior-posterior), arrowed lines represent Sagittal measurements.

calculated using (maximum AP dimension) x (maximum longitudinal dimension) [both placed on the mid-sagittal T2W image] x (maximum transverse dimension) [placed on the axial T2W image] x 0.52 [18]. MRI volume was additionally calculated using whole gland segmentation software (DynaCAD Prostate™, Philips). All US and MRI-derived measurements were performed by two experienced uroradiologists independently, and blinded to any clinical details.

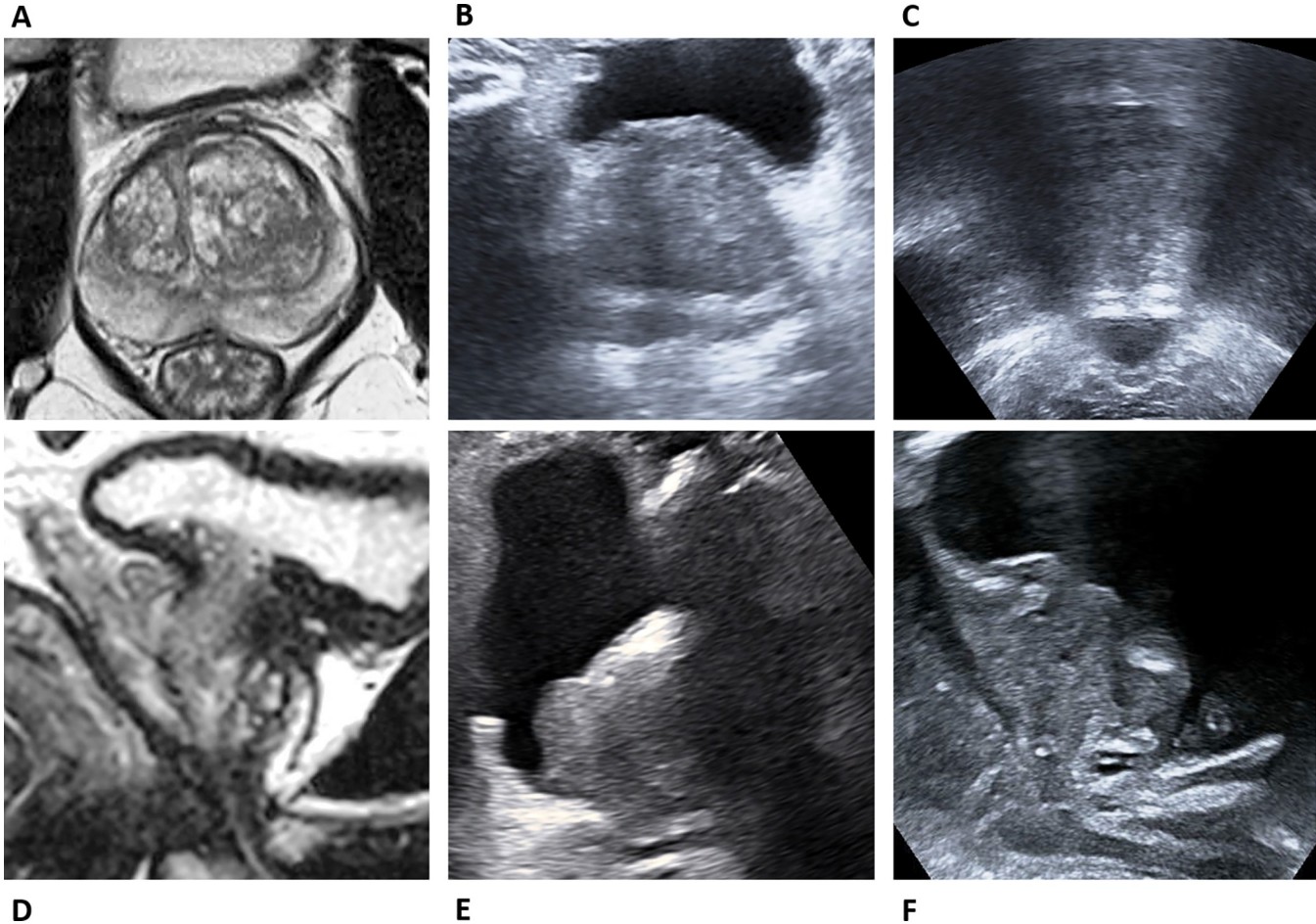

**Fig 2. Example Transabdominal (TA) and Transperineal (TP) US views.** Axial (A) and sagittal (D) MR images. B, E: TA ultrasound views from the same patient in the axial (B) and sagittal planes (E). C, F: TP ultrasound views in the axial (C) and sagittal planes (F). The sagittal TA US view (E) has shadowing from the pubis bone overlying the prostatic apex, which may affect measurement accuracy, conversely shadowing affects the axial view of TP imaging (C).

## Statistics

A two-way mixed effects model intra-class correlation coefficient (ICC) was used to measure how closely the US derived prostate volumes (TA, TP, TA/TP and TP-TA) compared to the gold-standard triplanar MRI-derived prostate volume and to assess inter-rater agreement [19]. ICC values were defined as indicating poor ($<0.5$), moderate (0.5–0.75), good (0.75–0.90) and excellent reliability ($>0.90$) [20]. We considered any cases with more than 50% percentage difference between the MRI- and any of the US- derived prostate volumes as outliers. Bland-Altman Agreement was also used to measure how closely the US derived prostate volumes (TA,

**Table 1. US views acquired for deriving prostate volumes.**

| View | Axial 1 (R-L) | Axial 2 (A-P) | Cranio-caudal |
|---|---|---|---|
| TA US | Transabdominal | Transabdominal | Transabdominal |
| TP US | Transperineal | Transperineal | Transperineal |
| TA/TP | Transabdominal | Transabdominal | Transperineal |
| TP-TA US | Transabdominal | Transperineal | Transperineal |

TP, TA/TP and TP-TA) compared to the gold-standard MRI-derived prostate volume. Categorical comparison was made between MRI PSA-density (MRI-PSAD) and each US-PSAD at three PSAD thresholds based on prior literature [14, 17, 21]: <0.15, 0.15–0.20, >0.20 ng/mL$^2$ in order to ascertain whether each US-PSAD was in agreement with the MRI-PSAD decision on risk stratification. In addition, a PSA-Categorical comparison was made between the MRI PSA-density and each of the four US derived PSA-densities at these PSAD cut-off values in order to ascertain whether each US-derived PSA density would be in agreement with the MRI derived PSA-density and the theoretical decision of whether to perform biopsy or not.

## Results

64 patients completed the study, with a mean age of 69 years (IQR: 64–73), mean PSA of 7.0 ng/mL (IQR: 5–9 ng/mL), and mean BMI of 27 (IQR: 24–29); **Table 2**. 36/64 (56.3%) patients had biopsy-confirmed prostate cancer: 18 Grade Group 1, 15 Grade Group 2, and three Grade Group ≥3.

### Prostate volumetric analysis

The mean volumetric segmentation-derived MRI prostate volume was 60 mL (IQR: 40–76 mL). The mean triplanar MRI-derived prostate volume was 61 mL (IQR: 39–79 mL). The average US derived volumes were 56 mL (IQR: 34–75 mL) for TA, 65 mL (46–71 mL) for TP, 66 mL (42–82 mL) for TA/TP, and 67 mL (46–80 mL) for TA-TP.

### ICC agreement between US and MRI-derived gland volumes

ICC revealed good agreement between radiologist-measured, clinical standard triplanar MRI-volume versus the gold-standard volumetric derived segmentation-derived MRI volumes (ICC 0.965; **Table 3**). Good agreement was demonstrated between all US derived prostate volumes and the gold-standard MRI-derived volumes calculated using 3-plane measurements, with highest ICC for US TA and US TA/TP Volume at 0.873 (95% CI: 0.798–0.921) and 0.874 (95% CI: 0.755–0.931), respectively. The lowest level of agreement was demonstrated for US TP Volume at 0.776 (95% CI: 0.654–0.858) (**Table 3**). A maximum of 3/64 cases lay outside the 95% confidence interval (CI) of agreement lines.

### ICC inter-rater agreement

Excellent inter-reader agreement was shown for readers' MRI and US-derived prostate volumes, with the greatest agreement noted between US-TA/TP and US TA volumes at 0.977 (95% CI: 0.959–0.987) and 0.963 (0.916–0.981), respectively (**Table 4**). 6/64 cases showed ≥50% percentage difference between the MRI-derived volume and at least one of the US-

**Table 2. Patient demographics.**

| Characteristic | |
|---|---|
| Patient age, years (IQR) | 69 (64–73) |
| PSA level, ng/mL (IQR) | 7.0 (5.0–9.0) |
| BMI (IQR) | 27.0 (24.0–29.0) |
| *Biopsy outcome* | |
| Negative | 28 (43.8%) |
| Gleason 3+3 | 18 (28.1%) |
| Gleason 3+4 | 15 (23.4%) |
| Gleason ≥4+3 | 3 (4.7%) |

**Table 3. Intraclass correlation coefficient (ICC) agreement between MRI-derived and US-derived prostate volumes.**

| Analysis | ICC (95% CI) |
|---|---|
| 3-plane MRI vs MRI Volumetric (Dynacad) | 0.965 (0.942–0.978) |
| 3-plane MRI vs US TA Volume | 0.873 (0.798–0.921) |
| 3-plane MRI vs US TP Volume | 0.776 (0.654–0.858) |
| 3-plane MRI vs US TA/TP Volume | 0.874 (0.755–0.931) |
| 3-plane MRI vs TA-TP Volume | 0.847 (0.709–0.915) |

derived calculations and could be considered as relative outliers. Out of these, two patients had a BMI of >30, with one further patient noted to have a defect due to a previous transurethral resection of the prostate (TURP) procedure, limiting the accuracy of the segmentation-derived volume (**Figs 3 and 4**).

## Bland Altman agreement between US and MRI-derived gland volumes

Good agreement was demonstrated between all US derived prostate volumes and gold-standard MRI-derived volumes calculated using whole gland segmentation software. A maximum of 3/64 cases lay outside the 95% confidence interval (CI) of agreement lines, with the highest agreement demonstrated between MRI and US-derived prostate volumes using either combined TA-TP or transperineal alone volumes (**S1 Fig**). Good inter-reader agreement was shown for readers' MRI and US-derived prostate volumes, with the greatest agreement noted between US-TP and US TP-TA volumes. Poorest agreement was demonstrated for the US-TA volumes (**S2 Fig**).

## Categorical comparison between MRI and US derived PSA-density

When evaluating the three PSA-density categories of $<0.15$ ng/mL$^2$, $0.15$–$0.20$ ng/mL$^2$ and $>0.20$ ng/mL$^2$, agreement was demonstrated between triplanar-derived MRI-PSAD and US-PSAD in 83–88% of cases, with highest agreement for TA/TP US-PSAD (**Table 5**). Simplifying this to a decision to biopsy or not at a PSA-density threshold of $>0.15$ ng/mL$^2$, US PSAD matched the MR decision in 86–91% of cases with the TP-TA US-PSAD performing best. Given that the ultrasound-derived measurements tended to over-estimate prostatic volume, we applied a lower, and clinically more conservative threshold of $0.12$ ng/mL$^2$ for a decision to proceed to biopsy. This resulted in an increased agreement with the MRI-equivalent decision of 92–95% of cases with highest agreement for transabdominal US-PSAD (**Table 5**).

Sensitivity, specificity, positive and negative predictive values were calculated for a matched decision to proceed to biopsy or not based on MRI PSAD and each U/S-derived PSAD at thresholds of $>0.15$ ng/mL$^2$ (**Table 6**) and $> 0.12$ ng/mL$^2$ (**Table 7**) respectively. At a biopsy threshold of $>0.15$ ng/mL$^2$, sensitivity ranged between 74–78% and specificity 98%-100%, PPV ranged between 96%-100% and NPV 88%-100%, with the optimal results obtained by the

**Table 4. Intraclass correlation coefficient (ICC) inter-observer agreement for MRI- and US- derived prostate volumes.**

| Analysis | ICC (95% CI) |
|---|---|
| 3-plane MRI Volume | 0.984 (0.974–0.99) |
| US TA Volume | 0.963 (0.916–0.981) |
| US TP Volume | 0.903 (0.84–0.941) |
| US TA/TP Volume | 0.977 (0.959–0.987) |
| US TA-TP Volume | 0.913 (0.856–0.947) |

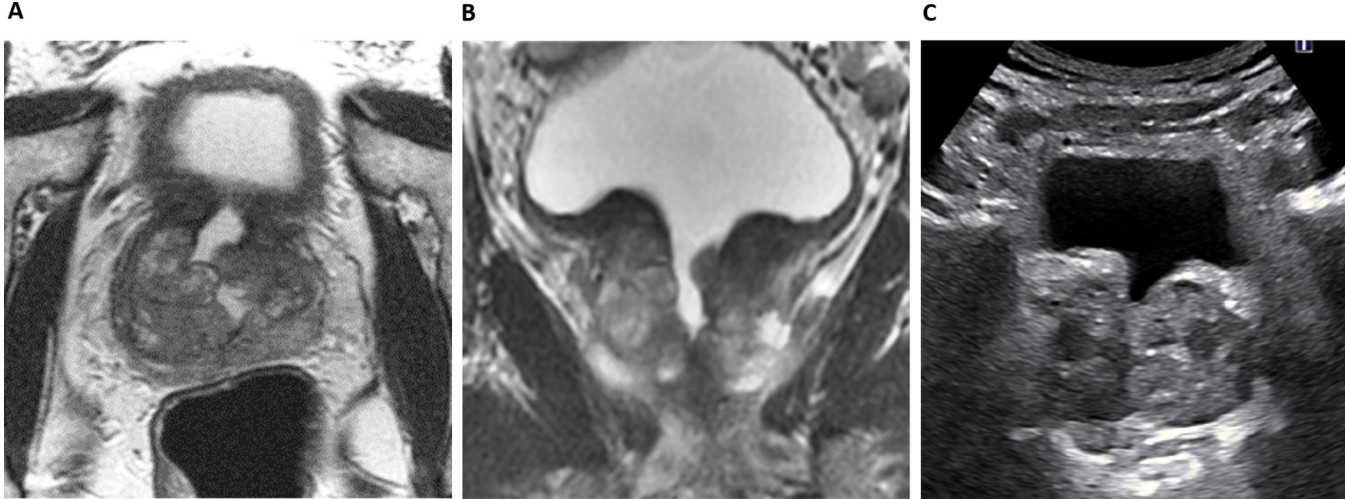

**Fig 3. TURP defect affects measurements.** Axial (A) and coronal (B) T2-weighted images show TURP defect at the base of the gland, with corresponding transverse transabdominal US image (C). MRI triplanar and segmentation volumes will differ, and sagittal US measurements will vary depending if the gland is measured in the midline position, or more laterally.

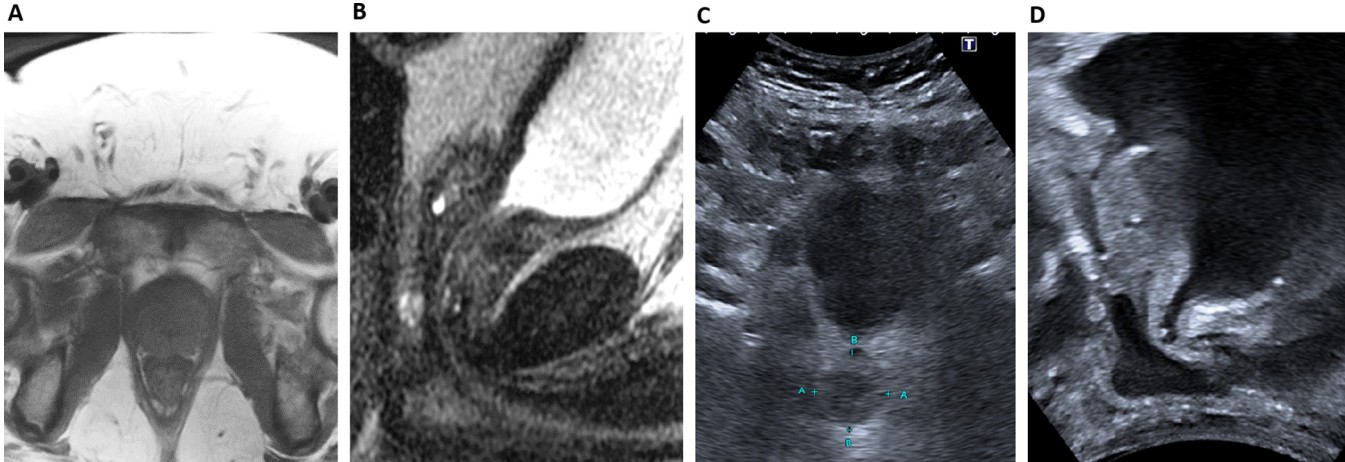

**Fig 4. Limited views due to patient body habitus.** MRI Axial T1-weighted image (A) shows depth of prostate; corresponding sagittal T2-weighted image of the prostate (B). C: Transabdominal transverse imaging is limited (calipers outline prostate). D: Transperineal sagittal views more clearly demonstrate the prostate.

**Table 5. Categorical comparison of MRI and US-derived PSA-density (PSA-D) at different biopsy thresholds of $<0.15$ ng/mL$^2$, 0.15–0.20 ng/mL$^2$ and $>0.20$ ng/mL$^2$) and agreement to biopsy or not at PSA-density thresholds of 0.15 ng/mL$^2$ and 0.12 ng/mL$^2$.**

|  | US-agreement with MRI volume at PSA-D categories | US-MRI agreement at PSA-D threshold 0.15 ng/mL$^2$ | US-MRI agreement at PSA-D threshold 0.12 ng/mL$^2$ |
|---|---|---|---|
| TA US-PSA density | 53/64 (83%) | 55/64 (86%) | 61/64 (95%) |
| TP US-PSA density | 54/64 (84%) | 56/64 (88%) | 60/64 (94%) |
| TA/TP US-PSA density | 56/64 (88%) | 56/64 (88%) | 59/64 (92%) |
| TP-TA US-PSA density | 54/64 (84%) | 58/64 (91%) | 59/64 (92%) |

**Table 6.  Sensitivity, Specificity, Positive predictive value (PPV), Negative predictive value (NPV) between MRI-PSA density and each US PSA density for the decision to biopsy at PSA threshold >0.15 ng/mL$^2$.**

|  | Sensitivity | Specificity | PPV | NPV |
|---|---|---|---|---|
| **TA US-PSA density** | 100% | 98% | 96% | 100% |
| **TP US-PSA density** | 91% | 100% | 100% | 95% |
| **TA/TP US-PSA density** | 74% | 100% | 100% | 88% |
| **TP-TA US-PSA density** | 78% | 100% | 100% | 89% |

US TA views, with a sensitivity of 100% and specificity 98%. At a biopsy threshold of >0.12 ng/mL$^2$ sensitivity ranged between 81–100%, specificity 85–100%, PPV 86–100% and NPV between 83–100%. Prioritizing sensitivity, US TA had optimal sensitivity of 100%, with a specificity of 85%, and when maximizing specificity, US TP was optimal with sensitivity of 87% and specificity 100%.

## Discussion

Our results demonstrate good overall agreement between surface-based US-derived and prostate volumes compared to MRI measurements with the greatest agreement shown between TA and TA/TP US views and MRI volume. We also demonstrate that risk-stratified decisions to biopsy based on US-derived PSAD thresholds strongly agree with MRI-derived measurements and therefore show potential for use as an initial point-of-care clinical test for either triaging MRI urgency at higher PSAD thresholds, or for avoiding the need for further investigations at lower thresholds. The results are encouraging as US is relatively inexpensive, readily available and non-invasive, and transabdominal scanning is a core sonographic skill [22].

Digital rectal examination estimated prostate size is known to lack accuracy, significantly underestimating prostate volume, particularly for larger glands, and cannot be used to derive measurements clinically [23]. MRI has been shown to be more accurate in volume assessment compared to transrectal US (TRUS) volumes estimated by urologists at the time of biopsy [24]. However, to our knowledge, no study has yet assessed PSA-density derived from the less invasive transabdominal or transperineal US approaches, and when performed by experienced radiologists. Ozden, et al compared transabdominal US and TRUS sagittal, transverse and AP prostatic diameters, finding a strong correlation between the two techniques, with the greatest correlation demonstrated for sagittal views [25]. Likewise, Pate, et al found that prostate transabdominal volume showed good agreement with TRUS volume and could be used interchangeably when the prostate volume was less than 30 mL, however, for larger glands, the authors recommend cross-sectional imaging for more accurate assessment [26]. Moreover, a recent systematic review comparing TRUS, CT and MRI prostatic volume measurements showed MRI as the optimal technique for deriving prostate volume [27]. Paterson, et al confirm these findings by demonstrating a better correlation of MRI prostate volume than TRUS using prostatectomy as the gold standard [24].

**Table 7.  Sensitivity, Specificity, Positive predictive value (PPV), Negative predictive value (NPV) between MRI-PSA density and each US PSA density for the decision to biopsy at PSA threshold >0.12 ng/mL$^2$.**

|  | Sensitivity | Specificity | PPV | NPV |
|---|---|---|---|---|
| **TA US-PSA density** | 100% | 85% | 86% | 100% |
| **TP US-PSA density** | 87% | 100% | 100% | 89% |
| **TA/TP US-PSA density** | 84% | 100% | 100% | 87% |
| **TP-TA US-PSA density** | 81% | 100% | 100% | 83% |

Despite its clinical utility, pre-biopsy prostate MRI is experience and quality dependent [28–30] and has ongoing limitations, leading to unnecessary, negative biopsies in around a third of patients [9–12], missing up to 20% of clinically significant cancers, and over-detecting insignificant lesions in around 10% of patients [13, 31–33]. However, MRI-derived PSA-density, which adjusts the PSA levels proportionate to gland size is emerging as a useful clinical biomarker, with a threshold of 0.15 ng/mL$^2$ suggested as a diagnostic cut-off that could be used to avoid help biopsy in selected patients [17]. MRI is an expensive and relatively scarce resource, therefore rationalizing use of such imaging would be of great clinical benefit. We investigated the agreement between MRI-PSA density and US-derived PSA density at the clinically meaningful threshold of >0.15 ng/ml$^2$ and subsequently the more conservative threshold of >0.12 ng/ml$^2$, also in part informed by the trend towards volume over-estimation with US. The results suggest that US-derived PSA density-based decisions for biopsy match the MRI-derived decision most closely with US-TA views at a biopsy threshold of >0.15 and with either TP or TA views at a biopsy threshold of >0.12 depending whether sensitivity of specificity is maximized. Higher sensitivity is likely to be more of a clinical priority for decisions to biopsy, and this is provided by transabdominal views alone, likely due to the propensity to under-estimate gland volume and therefore "falsely increase" PSA density. However, transperineal views either alone or in combination with transabdominal views may be of benefit in certain clinical circumstances, because the views are relatively independent of the degree of bladder filling and the body habitus of the patient. Transabdominal US provides excellent views in the axial plane, however, sagittal measurements may be compromised by shadowing from the pubic bones at the level of the prostatic apex (**Fig 2**), conversely transperineal US sagittal views are unaffected by this when scanning across the symphysis pubis, however, the right-left axial measurements are significantly compromised by pubic bone shadowing, leading to an overall poorer performance.

Our study has some limitations, including the relatively small sample size. We chose to compare US-derived measured to the clinical standard of triplanar MRI measurements rather than MRI segmentation derived prostate volumes. This matches current PI-RADS guidelines recommendations, given that segmentation can be time-consuming and software is not widely available [34]. Slight differences between triplanar and MRI volumetric measurements are not unexpected, as the ellipsoid formula assumes a regularly deformed sphere, however, asymmetrical BPH will lead to an over-estimation with this methodology, consistent with our findings [14, 35]. Further small differences may be observed if using PI-RADS version 2, rather than our chosen methodology of PI-RADS v2.1 [36]. TZ volume-derived may show better correlation to clinical outcomes compared to whole gland volume PSA-density [37], however, there is a close relationship between age-related increases in both [38], TZ volume is more challenging to quantify at US, and whole gland volumes are more widely accepted in clinical practice [17]. US image-acquisition quality is operator-dependent and, in particular for the more specialist technique of transperineal acquisition, performed by uro-radiologists; this may not be available in all healthcare settings, and may therefore affect the generalizability of our results. More studies are needed to evaluate whether these results can be replicated by non-specialist radiologists or sonographers, and whether there is potential for future translation into a primary care setting. Despite the prospective study design, the results of the study did not influence clinical decision-making. It should also be noted that in the clinical setting PSAD is used alongside MRI findings and therefore may be insufficient alone to avoid MRI, and prospective studies would be required to evaluate US-derived PSAD measurements against clinical outcomes and the decision to biopsy. It may also be possible to combine US-PSAD with blood or urine biomarkers, such as PHI to further augment a risk-stratified clinical decision process [39]. US did not always provide accurate measurements, however, we found this was often predictable, for

instance in patients undergoing a prior TURP procedure, or having a high BMI. It should be noted that post-TURP gland volumes may also be over-estimated by MRI segmentation software of triplanar measurements, however, this is not a limitation of MR imaging itself, rather the measuring technique employed.

In conclusion, we demonstrate that surface-based US-derived PSA density has good agreement with MRI-derived values and with the clinical risk-stratified decision for biopsy, and offers potential for supplementing clinical decision-making to triage patients to undergo urgent MR imaging or even avoid unnecessary MRI in selected patients.

## Supporting information

**S1 Fig. Bland-Altman agreement of US-derived TP versus MRI-derived prostate volume.** The middle solid line represents the mean agreement difference between MRI and US TP volumes, the lower line represents the lower limit of agreement between the two techniques ([mean– 1.96 standard deviation (SD) and the upper line represents the upper limit of agreement ([mean + 1.96 SD)] between the two techniques.
(TIF)

**S2 Fig. Bland-Altman agreement of US-derived TA versus MRI-derived prostate volume.** The middle solid line represents the mean agreement difference between MRI and US TP volumes, the lower line represents the lower limit of agreement between the two techniques ([mean– 1.96 standard deviation (SD) and the upper line represents the upper limit of agreement ([mean + 1.96 SD)] between the two techniques.
(TIF)

## Author Contributions

**Conceptualization:** Anne George, Vincent J. Gnanapragasam, Tristan Barrett.

**Data curation:** Maria Pantelidou, Iztok Caglic, Oleg Blyuss, Tristan Barrett.

**Formal analysis:** Maria Pantelidou, Iztok Caglic, Oleg Blyuss, Tristan Barrett.

**Funding acquisition:** Vincent J. Gnanapragasam.

**Investigation:** Anne George, Oleg Blyuss.

**Methodology:** Anne George.

**Resources:** Anne George.

**Software:** Oleg Blyuss.

**Supervision:** Vincent J. Gnanapragasam.

**Writing – original draft:** Maria Pantelidou, Tristan Barrett.

**Writing – review & editing:** Iztok Caglic, Anne George, Oleg Blyuss, Vincent J. Gnanapragasam, Tristan Barrett.

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
