## [Decision Letter · Decision Letter 0]

17 May 2022

PONE-D-22-06966Evaluation of transabdominal and transperineal ultrasound-derived prostate specific antigen (PSA) density and clinical utility compared to MRI prostate volumes: a feasibility studyPLOS ONE

Dear Dr. Barrett,

Thank you for submitting your manuscript to PLOS ONE. After assessing your work with the help of an expert reviewer, we  invite you to submit a revised version of the manuscript that addresses the points raised during the review process.

We look forward to receiving your revised manuscript.

Kind regards,

Pascal A. T. Baltzer, M.D.

Academic Editor

PLOS ONE

Journal Requirements:

“This research was supported by the National Institute of Health Research Cambridge Biomedical Research Centre (BRC-1215-20014). The views expressed are those of the author(s) and not necessarily those of the NIHR or the Department of Health and Social Care. The authors also acknowledge support from Cancer Research UK (Cambridge Imaging Centre grant number C197/A16465), the Engineering and Physical Sciences Research Council Imaging Centre in Cambridge and Manchester and the Cambridge Experimental Cancer Medicine Centre.”

“This research was supported by the National Institute of Health Research Cambridge Biomedical Research Centre (BRC-1215-20014). The views expressed are those of the author(s) and not necessarily those of the NIHR or the Department of Health and Social Care. The authors also acknowledge support from Cancer Research UK (Cambridge Imaging Centre grant number C197/A16465), the Engineering and Physical Sciences Research Council Imaging Centre in Cambridge and Manchester and the Cambridge Experimental Cancer Medicine Centre.

4. We note you have included a table to which you do not refer in the text of your manuscript. Please ensure that you refer to Table 6 and 7 in your text; if accepted, production will need this reference to link the reader to the Table.

5. Please include your tables as part of your main manuscript and remove the individual files. Please note that supplementary tables (should remain/ be uploaded) as separate "supporting information" files.

Reviewers' comments:

Reviewer's Responses to Questions

**Comments to the Author**

1. Is the manuscript technically sound, and do the data support the conclusions?

Reviewer #1: Yes

2. Has the statistical analysis been performed appropriately and rigorously? 

Reviewer #1: Yes

3. Have the authors made all data underlying the findings in their manuscript fully available?

Reviewer #1: No

4. Is the manuscript presented in an intelligible fashion and written in standard English?

Reviewer #1: Yes

5. Review Comments to the Author

Reviewer #1: In the present study, the authors evaluate the accuracy of surface-based ultrasound-derived PSAD metrics compared to MRI. The patients were prospectively recruited, but no prospective clinical decision making was performed based on the novel data. The authors used MRI-ellipsoid-derived measurements rather than the volumetric segmentation in order to better reflect the clinical routine. The study is well written and a pleasure to read. The results are presented in a concise fashion. For a first small pilot study, it is methodologically sound. I only have a few minor points which should be addressed prior to publication:

1. In accordance with PLOS ONE policy, the authors should make their raw data available with this study. At least, this should be a CSV file containing all the measurements and volumes of all subject. Though sharing the image data on a public repository such as XNAT is usually desirable, it may not be needed for a small pilot study such as this one.

2. Could the authors explicitly specify how the triplanar measurement was performed on MRI? (presumably according to PI-RADS 2.1 recommendations?) This has a small influence on the volume estimate (see Ghafoor et al., 10.1016/j.acra.2020.07.027)

3. The authors could add Bland-Altman plots, which would make it easy for the reader to visually grasp the disagreement between the methods (residual error may be heteroscedastic)

4. If the authors see any additional value in it, they could either just discuss or even perform post hoc estimation of the TZ volume and TZPSAD, which may be slightly superior to whole gland PSAD (see Schneider et al, 10.1016/j.ejrad.2019.108660).

5. Re "limiting the accuracy of the segmentation-derived volume" [Results; p14]: This is rather an expression of the limited functionality of the software used than a true limitation of MRI. TURP defects are well delineated on T2w imaging. With state-of-the-art software such as ImageJ, accurate segmentations can still be provided.

6. PLOS authors have the option to publish the peer review history of their article (what does this mean?). If published, this will include your full peer review and any attached files.

Reviewer #1: **Yes: **Anton S. Becker, MD, PhD

---

## [Author Response · Author response to Decision Letter 0]

25 May 2022

“This research was supported by the National Institute of Health Research Cambridge Biomedical Research Centre (BRC-1215-20014). The views expressed are those of the author(s) and not necessarily those of the NIHR or the Department of Health and Social Care. The authors also acknowledge support from Cancer Research UK (Cambridge Imaging Centre grant number C197/A16465), the Engineering and Physical Sciences Research Council Imaging Centre in Cambridge and Manchester and the Cambridge Experimental Cancer Medicine Centre.”

We have removed this from the Acknowledgements section. The funding section included all relevant funding and has been left unchanged.

We have removed tis text from the manuscript.

“This research was supported by the National Institute of Health Research Cambridge Biomedical Research Centre (BRC-1215-20014). The views expressed are those of the author(s) and not necessarily those of the NIHR or the Department of Health and Social Care. The authors also acknowledge support from Cancer Research UK (Cambridge Imaging Centre grant number C197/A16465), the Engineering and Physical Sciences Research Council Imaging Centre in Cambridge and Manchester and the Cambridge Experimental Cancer Medicine Centre.

Please see above. No change required.

We have uploaded the dataset file on https://data.mendeley.com/, with the appropriate original data file relevant to this study (DOI:10.17632/2bvdcspxx2.1); it can be accessed via https://data.mendeley.com/datasets/2bvdcspxx2/1 

We have uploaded the dataset file on https://data.mendeley.com/, with the appropriate original data file relevant to this study.

Not applicable. 

4. We note you have included a table to which you do not refer in the text of your manuscript. Please ensure that you refer to Table 6 and 7 in your text; if accepted, production will need this reference to link the reader to the Table.

The reference to the table had been combined within the text as “ (Tables 6, 7) ”. For added clarity we have changed this to:

“PSAD at thresholds of >0.15 ng/mL2 (Table 6) and > 0.12 ng/mL2 (Table 7), respectively.”

5. Please include your tables as part of your main manuscript and remove the individual files. Please note that supplementary tables (should remain/ be uploaded) as separate "supporting information" files.

The change has been made.

No changes have been made, other than in response to the reviewer’s comments – the relevant changes to the reference are apparent on the tracked version of the resubmitted document.

Reviewers' comments:

Reviewer's Responses to Questions

Comments to the Author

1. Is the manuscript technically sound, and do the data support the conclusions?

Reviewer #1: Yes

2. Has the statistical analysis been performed appropriately and rigorously?

Reviewer #1: Yes

3. Have the authors made all data underlying the findings in their manuscript fully available?

Reviewer #1: No

We have uploaded the dataset file on https://data.mendeley.com/, with the appropriate original data file relevant to this study.

4. Is the manuscript presented in an intelligible fashion and written in standard English?

Reviewer #1: Yes

5. Review Comments to the Author

Reviewer #1: In the present study, the authors evaluate the accuracy of surface-based ultrasound-derived PSAD metrics compared to MRI. The patients were prospectively recruited, but no prospective clinical decision making was performed based on the novel data. The authors used MRI-ellipsoid-derived measurements rather than the volumetric segmentation in order to better reflect the clinical routine. The study is well written and a pleasure to read. The results are presented in a concise fashion. For a first small pilot study, it is methodologically sound. I only have a few minor points which should be addressed prior to publication:

1. In accordance with PLOS ONE policy, the authors should make their raw data available with this study. At least, this should be a CSV file containing all the measurements and volumes of all subject. Though sharing the image data on a public repository such as XNAT is usually desirable, it may not be needed for a small pilot study such as this one.

Please see above, we have uploaded the dataset file on https://data.mendeley.com/, with the appropriate original data file relevant to this study (DOI:10.17632/2bvdcspxx2.1); it can be accessed via https://data.mendeley.com/datasets/2bvdcspxx2/1 

2. Could the authors explicitly specify how the triplanar measurement was performed on MRI? (presumably according to PI-RADS 2.1 recommendations?) This has a small influence on the volume estimate (see Ghafoor et al., 10.1016/j.acra.2020.07.027)

This was following the PI-RADS version 2.1 guidelines ellipsoid formulation. We have clarified this in the methodology as below, and used the relevant PI-RADS v2.1 reference.

“…following the PI-RADS version 2.1 guidelines ellipsoid formulation; calculated using (maximum AP dimension) x (maximum longitudinal dimension) [both placed on the mid-sagittal T2W image] x (maximum transverse dimension) [placed on the axial T2W image] x 0.52 [Ref].”

We have also mentioned the potential for differences between v2.0 and v2.1 in the limitations section and referenced the suggested paper.

3. The authors could add Bland-Altman plots, which would make it easy for the reader to visually grasp the disagreement between the methods (residual error may be heteroscedastic)

We have now performed a Bland-Altman Agreement, detailed in the methods and results with representative supplemental figures uploaded. Added:

Methods:

“Bland-Altman Agreement (BAA) was also used to measure how closely the US derived prostate volumes (TA, TP, TA/TP and TP-TA) compared to the gold-standard MRI-derived prostate volume.”

Results:

“Good agreement was demonstrated between all US derived prostate volumes and gold-standard MRI-derived volumes calculated using whole gland segmentation software. A maximum of 3/64 cases lay outside the 95% confidence interval (CI) of agreement lines, with the highest agreement demonstrated between MRI and US-derived prostate volumes using either combined TA-TP or transperineal alone volumes (Supplemental Figure 1). Good inter-reader agreement was shown for readers’ MRI and US-derived prostate volumes, with the greatest agreement noted between US-TP and US TP-TA volumes. Poorest agreement was demonstrated for the US-TA volumes (Supplemental Figure 2).”

4. If the authors see any additional value in it, they could either just discuss or even perform post hoc estimation of the TZ volume and TZPSAD, which may be slightly superior to whole gland PSAD (see Schneider et al, 10.1016/j.ejrad.2019.108660).

While TZ volume may better show better correlation, both whole gland (WG) volume and TZ volume show a similar increase with age (Bura, et al Eur Radiol. 2021;31(7):4908-4917), allowing for the confounding factor of an enlarged TZ variably causing compression of the PZ to differing degrees between patients. Furthermore, the TZ cannot be reliably assessed with Ultrasound (especially retrospectively in this cohort), and WG derived PSA-density is currently more widely used in clinical practice. We have added these references and paraphrased this discussion within the limitations. 

5. Re "limiting the accuracy of the segmentation-derived volume" [Results; p14]: This is rather an expression of the limited functionality of the software used than a true limitation of MRI. TURP defects are well delineated on T2w imaging. With state-of-the-art software such as ImageJ, accurate segmentations can still be provided.

The MRI volumetric measurements were considered accurate and rather it was the US measurements that were problematic, however, the reviewer’s comment is noted and we have added this to the discussion section.

6. PLOS authors have the option to publish the peer review history of their article (what does this mean?). If published, this will include your full peer review and any attached files.

Do you want your identity to be public for this peer review? For information about this choice, including consent withdrawal, please see our Privacy Policy.

Reviewer #1: Yes: Anton S. Becker, MD, PhD

---

## [Decision Letter · Decision Letter 1]

22 Aug 2022

Evaluation of transabdominal and transperineal ultrasound-derived prostate specific antigen (PSA) density and clinical utility compared to MRI prostate volumes: a feasibility study

PONE-D-22-06966R1

Dear Dr. Barrett,

We’re pleased to inform you that your manuscript has been judged scientifically suitable for publication and will be formally accepted for publication once it meets all outstanding technical requirements.

Kind regards,

Pascal A. T. Baltzer, M.D.

Academic Editor

PLOS ONE

**Comments to the Author**

1. If the authors have adequately addressed your comments raised in a previous round of review and you feel that this manuscript is now acceptable for publication, you may indicate that here to bypass the “Comments to the Author” section, enter your conflict of interest statement in the “Confidential to Editor” section, and submit your "Accept" recommendation.

Reviewer #1: All comments have been addressed

2. Is the manuscript technically sound, and do the data support the conclusions?

Reviewer #1: Yes

3. Has the statistical analysis been performed appropriately and rigorously? 

Reviewer #1: Yes

4. Have the authors made all data underlying the findings in their manuscript fully available?

Reviewer #1: Yes

5. Is the manuscript presented in an intelligible fashion and written in standard English?

Reviewer #1: Yes

6. Review Comments to the Author

Reviewer #1: All comments have been addressed. Agree with the reply. Congratulations on this well written manuscript!

7. PLOS authors have the option to publish the peer review history of their article (what does this mean?). If published, this will include your full peer review and any attached files.

Reviewer #1: **Yes: **Anton S. Becker, MD, PhD

---

## [Editor Report · Acceptance letter]

1 Sep 2022

PONE-D-22-06966R1 

Evaluation of transabdominal and transperineal ultrasound-derived prostate specific antigen (PSA) density and clinical utility compared to MRI prostate volumes: a feasibility study 

Dear Dr. Barrett:

I'm pleased to inform you that your manuscript has been deemed suitable for publication in PLOS ONE. Congratulations! Your manuscript is now with our production department. 

Kind regards, 

on behalf of

Dr. Pascal A. T. Baltzer 

Academic Editor

PLOS ONE